# A Study of the Variability of Volume Transport of the Benguela Current

Sudip Majumder[1] and Claudia Schmid[2]

[1]Cooperative Institute for Marine and Atmospheric Studies, University of Miami, Miami, Florida, USA.
[2]Atlantic Oceanographic and Meteorological Laboratory, NOAA, Miami, Florida, USA.

*Correspondence to:* Sudip Majumder (sudip.majumder@noaa.gov)

**Abstract.** The Benguela Current forms the eastern limb of the subtropical gyre in the South Atlantic and transports a blend of relatively fresh and cool Atlantic water as well as relatively warm and salty Indian Ocean water northwestward. Therefore, it plays an important role not only for the local freshwater and heat budgets but for the overall meridional heat and freshwater transports in the South Atlantic. Historically, the Benguela Current region is relatively data sparse, especially with respect to

5   long-term velocity observations. A new three dimensional data set of the horizontal velocity in the upper 2000 m that covers the years 1993 to 2015 is used to analyze the variability of the Benguela Current. This data set was derived using observations from Argo floats, satellite sea surface height and wind fields. Since Argo floats do not cover regions shallower than 1000 m, the dataset has gaps inshore. The main features of the horizontal circulation observed in this data set are in good agreement with those from earlier studies based on limited observations. Therefore, it can be used for a more detailed study of the flow pattern

10   as well as the variability of the circulation in this region. It is found that the mean meridional transport in the upper 800 m between the continental shelf of Africa and 3$^o$E, decreases from 23$\pm$3Sv (1 $Sv = 10^6 m^3/sec$) at 31$^o$S to 11$\pm$3Sv at 28$^o$S.

    In terms of variability, the 23-year long timeseries at 30$^o$S and 35$^o$S reveal phases with large energy densities at periods of 3 to 7 months, which can be attributed to the occurrence of Agulhas rings in this region. The prevalence of Agulhas rings is also behind the fact that the energy density at 35$^o$S at the annual period is smaller than at 30$^o$S, because the former latitude is closer

15   to Agulhas retroflection and therefore more likely to be impacted by the Agulhas rings. In agreement with this, the energy density associated with mesoscale variability at 30$^o$S is weaker than at 35$^o$S. With respect to the forcing, the Sverdrup balance and the observed transport at 30$^o$S exhibit strong correlation of 0.7. No significant correlation between these parameters is found at 35$^o$S.

# 1 Introduction

The broad nortwestward flow following the west coast of southern Africa from Cape Agulhas ($35^o$S, Figure 1) to Cape Frio ($18.4^o$ S, Figure 1) (Garzoli and Gordon, 1996) is the Benguela Current which constitutes the eastern limb of the south Atlantic sub-tropical gyre. The Benguela Current transports water masses carried into the Cape Basin by the South Atlantic Current, the Antarctic Circumpolar Current and the Agulhas Current (Gordon et al., 1992). The contribution from the Agulhas Current consists of warm, salty Indian Ocean water that enters the Atlantic in the Agulhas Retroflection Region via the shedding of rings and the Agulhas leakage (Lutjeharms and Van Ballegooyen, 1988). Agulhas rings are large (with diameter 300-400 km) and extremely energetic (Olson and Evans, 1986), in fact, the ring shedding region is characterized by significantly higher level of eddy kinetic energy than observed in the other parts of the world ocean in the southern hemisphere (Ducet et al., 2000). On average, the Agulhas rings transfer about 10-15 Sv (1 $Sv = 10^6 m^3/sec$) into the Atlantic in the upper 1000 m (Peterson and Stramma, 1991). Because of their water mass characteristics they are important for the heat and freshwater budget in the South Atlantic.

From its origin in the Cape Cauldron (Boebel et al., 2003) the Benguela Current flows northwestward along the west coastline of Africa and feeds into the southern South Equatorial Current (Stramma, 1991) which flows in a westerly direction between $8^o$S and $22^o$S (Rodrigues et al., 2007). At intermediate depth, the flow towards South America is more zonal, and once it reaches the boundary in the Santos Bifurcation (Boebel et al., 2003), about two thirds of the intermediate water contribute to the Brazil Current and one third to the northward flowing Intermediate Western Boundary Current (Schmid et al., 2000; Boebel et al., 2003).

The Benguela Current plays a key role for the Atlantic Meridional Overturning Circulation (AMOC) by transporting heat and salt from the Indian Ocean northwards. The AMOC is important for the global energy budget and is believed to be linked with multiple regional and global climate phenomenon as well as extreme weather events in the North America and around the globe (Sloyan and Rintoul, 2001; Haarsma et al., 2005; Garzoli and Matano, 2011; Lopez et al., 2016).

Recent model-based studies suggested that heat transfer to the North Atlantic by the Benguela Current increased due to an increase of the Indian Ocean inflow through the Agulhas leakage in 1965 to about 1990 (Biastoch et al., 2009, 2015). This increase was attributed to a strengthening of the Agulhas Current (Biastoch et al., 2009) because of a poleward shift of Southern hemisphere westerlies as reported in many studies based on climate models (Cai, 2006; Yang et al., 2016; Saenko et al., 2005). An increase in the Indian Ocean inflow could result in increased heat and salt transports into the South Atlantic causing salinification there, which could gradually extend into the North Atlantic (Biastoch et al., 2009). From about 1990 on, no significant change of the Agulhas leakage was detected by Biastoch et al. (2015) (their Figure 4). In agreement with this, a recent observational study by Beal and Elipot (2016) found that the Agulhas Current has not strengthened during the period 1993 to 2015. Instead, it has been broadening. Beal and Elipot (2016) observed that intensifying winds strengthen the eddy kinetic

energy of the Agulhas Current, but do not increase its mean transport.

As part of an early effort (Benguela Sources and Transport Experiment, BEST) based on direct current measurement with a moored current meter array and inverted echo sounders Garzoli and Gordon (1996) derived a northward transport of about 13 Sv ($1Sv=10^6$ m$^3$/sec) across $30^o$ S, between the Walvis Ridge in the west and the African coast in the east (Figure 1). They observed that 50 % of the meridional transport in the upper 1000 m at this latitude consists of waters from the South Atlantic, 25% from the Indian Ocean and the remaining 25% is a mix of water from the Indian and the tropical Atlantic Ocean. In addition, Garzoli and Gordon (1996) reported that the Benguela Current at $30^o$S between the African coast and $8^o$E consists of a relatively steady northwestward flow. Further west, they found that the flow becomes transient between $8^o$E and the Walvis Ridge at $3^o$E due to the influence of Agulhas rings. Complementing the BEST program a Benguela Current Float Experiment was conducted during 1997 - 1999 in an attempt to directly measure the northward flow of intermediate water using Lagrangian RAFOS floats in conjunction with moored sound sources and CTD/O2/LADCP profiles (Richardson and Garzoli, 2003). These observations suggested that the Benguela Current Extension (also called southern South Equatorial Current) in the upper 750 m is located between $35^o$S and $20^o$S. They reported that the westward transport of intermediate water in this current was about 15 Sv between $22^o$S and $35^o$S.

Even though the Benguela Current region features interesting physical processes, constitutes the eastern limb of the Meridional Overturning Circulation in the South Atlantic, and has an impact on an important up-welling region near the African Coast with high biological productivity, no long-term measurement of this current's flow/transport/dynamics are available.

In this study, using extensive observations from Argo and altimetry, we provide a 23 year long time series as well as the means of the transport of the Benguela Current. This data set provides horizontal velocities from observations in the upper 2000 m at a higher resolution in space and time than was available in earlier studies. Based on this data set, the variability of the Benguela Current from seasonal to interannual scales is analyzed. The results from this study will improve the knowledge of the flow patterns and will be helpful for model validation in this region. The primary goal of this manuscript is to improve the understanding of the variability of the Benguela Current transport between $25^o$S and $35^o$S.

## 2   Data and Methods

The methodology and the details of the product (called, Argo & SSH) are described in Schmid (2014). Improvements were implemented and the time series was extended in preparation for a study of the Meridional Overturning Circulation in the South Atlantic. Details about this can be found in Majumder et al. (2016). A short summary of the methodology for deriving the velocity fields follows:

(i) temperature and salinity profiles from Argo floats measured in the years 2000 to 2015, are used to calculate dynamic height;

(ii) to improve the monthly spatial data coverage, fits between sea surface height (SSH) from AVISO (AVISO, 1996) and the

dynamic heights are used to derive synthetic dynamic height profiles on a $0.5^o \times 0.5^o$ grid;

(iii) these synthetic dynamic height fields are used to calculate geostrophic velocities relative to a level of no motion;

(iv) absolute geostrophic velocity fields are obtained by adjusting the geostrophic velocity by using velocity fields obtained from the trajectories of subsurface floats;

(v) and, finally, the Ekman component, estimated from NCEP2 reanalysis winds (Kanamitsu et al., 2002), is added to the derived velocity fields.

The resulting timeseries is an extension of the one used by Majumder et al. (2016) and covers the years 1993 - 2015, of which the first seven fall in the pre-Argo period. This extension of the time series is based on the assumption that the relation-

ship between SSH and dynamic height does not change much over time (Dong et al., 2015; Lopez et al., 2017).

The derived gridded monthly velocity fields are used to estimate volume transports in the Benguela Current region. Wind stress curl and the Sverdrup stream functions are estimated using European Reanalysis interim wind fields (ERA interim) (Dee et al., 2011). This wind field has a 0.75-degree resolution and is available for the years 1979 to 2016.

**3   Results**

**3.1   Structure of the Benguela Current**

The northwestward flow along the southwestern coast of Africa is called the Benguela Current. Maps of the adjusted steric height by Reid (1989) show this current east of the Walvis Ridge. Following Reid (1989), at $30^o$S Garzoli and Gordon (1996) derived the transport of this current for the region between the eastern edge of the Walvis Ridge (at about $3^o$E) and western

African coast. For comparability with their transport estimates, the same region is used herein. The mean flow field in Figure 2 indicates that the northwestward flow of this current is east of $3^o$E between $35^o$S and $25^o$S. Therefore, for transport estimates the Benguela current is integrated zonally between $3^o$E and the African coast within this latitude range.

Climatologies of currents at 15 m and transports in the upper 800 m in Figure 2 clearly visualize the northwestward flowing Benguela Current and shows that it is fed by water from the southern Atlantic as well as the Indian Ocean. The former mostly

comes from the South Atlantic Current while the latter enters the South Atlantic via the Agulhas Retroflection. Between $35^o$S and $30^o$S the meridional velocity within the Benguela Current weakens as the current turns more westward (Figure 2a).

Two main regimes with different flow patterns are identified. The eastern regime near the African coast is characterized by relatively strong northward flow while the western regime has relatively weak alternating flow. For example, at $30^o$S the flow at 15 m west of $8^o$E is mostly zonal and meanders slightly. Farther south, the flow in the western regime does not reveal any

preferred direction. In addition to these regimes, south of $32^o$S reveals several distinct recirculation features in the mean field. One of them is in the white box with the center between the Walvis Ridge in the west and the Vema Seamount (at $33^o$S, $6^o$E, Figure 2) in the east.

The eastern regime with strong northward flow at 30$^o$S is broader than at 35$^o$S, extending from the African coast to about 8$^o$E (Figure 2). At 35$^o$S the mean 15 m velocity field indicates that the northward flow east of 12$^o$E may occur in two branches. The eastern one, between about 15$^o$E and the African coast, is dominated by meridional flow. In contrast to this, the western one, between 12$^o$E and 15$^o$E, has zonal velocity components that are as large as the meridional velocity. In addition, the magnitude of the velocity and transport are about twice as large as farther east. A third contributory branch can also be seen at about 7$^o$E. This branch, identified as part of a meander of the South Atlantic Current, has relatively weak contribution in the mean transport at 35$^o$S compared to the two prominent branches in the east.

The vertical structure of the climatological meridional velocity in the upper 1500 m in Figure 3 suggests that the northward flow east of 12$^o$E dominates throughout most of the upper 1500 m, especially at 35$^o$S. The exception of this is the southward Benguela Undercurrent near the eastern end of the section at 30$^o$S from about 600 m downward. The third contributory branch is seen between 5$^o$E to 7$^o$E, with relatively small northward velocities.

With respect to changes of the structure of the Benguela Current on its way to the north it can be seen that its strength decreases significantly in the eastern regime between 35$^o$S and 30$^o$S. In addition, one can see the two branches of the northward flow at 35$^o$S, distinguished by the factor two difference in the strength of the meridional velocity that are separated by the less deep-reaching flow near 15$^o$E. West of 12$^o$E at 35$^o$S, the velocity is mostly much smaller and of alternating sign when compared with the eastern regime (east of 12$^o$E). Also, the Benguela Current completed its westward turn at 30$^o$S which results in relatively weak meridional velocity and a meander-like pattern of the flow (Figure 2).

Before proceeding to zonal integrals of the transport it has to be noted that Argo & SSH does not contain velocities in boxes that are shallower than 1000 m at the center (as can be seen in Figure 3). One way to assess how much transport occurs in these shallow regions on average is to use the easternmost near-surface velocity at each latitude to derive an approximate transport in the regions that are missing. The average northward velocities of 5 cm/s for 30$^o$S and 10 cm/s for 35$^o$S yield 2.0 Sv and 1.8 Sv, respectively. These transports are about 10 % of the mean meridional transports at these two latitudes.

To compare with the estimates from previous studies [e.g.: Stramma and Peterson (1989); Garzoli and Gordon (1996)], this study calculates transport in the meridional direction; however, transport calculated along a section perpendicular to the flow is the same. The mean meridional transport of the Benguela Current in the upper 800 m ranges from 9$\pm$3 Sv to 23$\pm$3 Sv (Figure 4; whenever possible transports are represented as time-mean $\pm$ standard deviation, calculated over the entire time series). In the upper 1000 m, the range is 10$\pm$3 Sv to 26$\pm$3 Sv. The transports in the upper 1000 m are between 5% and 25% (1% and 5%) higher south (north) of 29$^o$S than those integrated over the upper 800 m. The agreement with estimates from previous studies is mostly good if one keeps in mind that most of them are for synoptic sections and use different vertical integration limits. In addition, their zonal integration limit in the west varies from Greenwich Meridian to 3$^o$E. For example Clement and Gordon (1995) and Stramma and Peterson (1989) both use Greenwich Meridian as the western edge of the Benguela Current at 32$^o$S,

where as Garzoli and Gordon (1996) integrated to 3$^o$E to obtain transport at 30$^o$S.

Between 35$^o$S and 31$^o$S, the transports are relatively stable when keeping the standard deviations in mind. From 31$^o$S to 28$^o$S the transports in the upper 800 m (1000 m) decrease from 23$\pm$3 Sv to 11$\pm$3 Sv (26$\pm$3 Sv to 12$\pm$3 Sv). This can be attributed to the westward turn of the flow as the Benguela Current feeds into the southern South Equatorial Current. North of 28$^o$S the transport are, once again relatively stable.

## 3.2 Transport Budget and its Uncertainties

To understand the circulation pattern and its variability in the Cape Basin region, volume transport budgets in the upper 800 m are derived for the boxes ABCD and A'BC'D (Figure 7). For the box ABCD, time series of the volume transport (Figure 5) yield mean northward transports of 18$\pm$3 Sv and 19$\pm$3 Sv across lines AB (at 30$^o$S) and CD (at 35$^o$S). More variable westward transports of 8$\pm$4 Sv and $\sim$0.3$\pm$3 Sv cross lines AC (at 3$^o$E) and BD (parallel to the shelf break). The latter is derived as the component of the transport perpendicular to the direction of the 1000 m isobath near the African coast. It will be called the cross-shelf transport, hereinafter. Combining transports across the sides of the box ABCD reveals an imbalance of 6 Sv in the upper 800 m within the box.

The previously discussed recirculation feature at the southwest corner of box ABCD in Figure 2 and a meandering flow with a northward component at the western end of line AB at 30$^o$S can impact the budget significantly. In addition to that, the topography near the Walvis Ridge can have an impact on the flow near the lower integration limit (800 m).

To understand the causes of this imbalance, the mean transport budget is estimated in the two boxes mentioned above and the integration depth in the larger box is varied. It is found that the mean transport imbalance in the upper 400 m for the box ABCD is about 2 Sv (4 Sv smaller than in the upper 800 m); the individual transports are about 12 Sv (northward) across lines AB and CD, about 4 Sv across AC (westward) and about 2 Sv across BD (westward). In the smaller box (A'BC'D), the mean transport in the upper 800 m is balanced, with individual transports of about 14 Sv (northward) across line A'B, 20 Sv (northward) across line C'D, about 7 Sv (westward) across A'C' and about 0.3 Sv (westward) across BD. These estimates indicate that about 4 Sv of the imbalance in the large box (ABCD) disappear when the vertical integration limit is reduced from 800 m to 400 m, which indicates that the Walvis Ridge gives rise to a topographic effect that reduces the chance of closing the budget. Consistent with this, retaining the full depth and shifting the western boundary eastward (away from the Walvis Ridge) to 8$^o$E (small box A'BC'D) results in a closed budget within the error bars.

Time variant transports including the budgets for ABCD and A'BC'D are presented in Figure 5. Even though, on an average, transport budget for the box A'BC'D seems to close, anomalies exist. Some of these exceed 5 Sv. The time series indicates that the most of the imbalances (Figure 5e, gray) occur at times with large anomalous cross-shelf transports (Figure 5d). In the

following the characteristics of three strongest events - occurring in July 1993, April 2005, and August 2007 - are explored.

Many studies (e.g. Shannon et al. (1983); Lutjeharms and Meeuwis (1987); Shannon (2006); Shannon et al. (2006); Hutchings et al. (2009)) showed that Ekman-induced upwelling plays an important role in the coastal circulation pattern east of the region dominated by the Benguela Current. Such upwelling typically occurs hand-in-hand with off-shore transports from the shelf into the open ocean. Shannon et al. (1983) identified several upwelling cells in the coastal Benguela regime. The important ones are the Cape Columbine and the Cape Peninsula cells between $30^oS$ and $35^oS$. Upwelling in these cells can modulate cross-shelf transport and consequently impact the transport budget. Based on the full time series, it is found that the correlation between the cross-shelf transport and the along-shelf wind stress from NCEP2 is low (0.14). The wind record is in poor agreement during strong events like the three listed above. This indicates that away from the coast at about 1000 m isobath oceanic processes that are independent of the wind have to give rise to these events. This is supported by a Hovmöller diagram of the monthly sea surface anomalies (Figure 7), which confirms the passage of eddies during each one of these three events. These eddies, located very close to the shelf break at $35^oS$, therefore, give rise to relatively large cross-shelf velocities (northeastward/southwestward) north of their center. Maps of sea surface height anomalies (not shown) during these events also confirm the strong influence of cyclones and anticyclones in the cross-shelf transport.

Other reasons of the anomalies can be:

(i) integration of the transport to the same depth limit for all sides of the box, (ii) transport at shallow depths near the African coast that are not represented in the velocity field, (iii) a vertical transport into the box from deeper layers, and (iv) a surface freshwater flux. These factors are discussed in the following.

It is found that both at $30^oS$ and at $35^oS$, $27 kg/m^3$ $\sigma_0$ isopycnal lies at a depth of 800 m $\pm$ 50 m (not shown). This indicates that the choice of 800 m as the depth limit for all three sides of the box cannot give rise to a significant cross-isopycnal transport. With respect to the impact of flow in shallow regions, as mentioned in the previous section, Argo & SSH misses about 2 Sv near the coast both at $30^oS$ and $35^oS$. Since the missed transports at the lines AB and CD are almost the same, they do not contribute to the imbalance.

Investigating the contribution due to a vertical transport through the bottom of the box can be done by approximating the Ekman transport using an upwelling velocity derived from the mean wind stress curl of $1.5 \times 10^{-7} N/m^3$ within the box. This velocity is $0.2 \times 10^{-3}$ cm/sec which corresponds to a transport of about 1.5 Sv. Because the Ekman depth is 65 m, the vertical transport at 800 m is likely to be smaller.

An estimate of the transport due to a surface freshwater flux is calculated using the climatological 'evaporation - precipitation' from European Center for Medium range Weather Forecasting (ECMWF) reanalysis. ECMWF's ERA-40 ($http://www.ecmwf.int/s/ERA-40\_Atlas/docs/section\_B/parameter\_emp.html$) has a climatological mean in the range

of 2 to 4 mm/day in the box that does not vary much from season to season. Based on a net surface freshwater flux of 3 mm/day (about 90 mm/month) the net transport into the box is 0.02 Sv. This transport is much smaller than the observed imbalance of 7 Sv. To achieve a gain of 1 Sv transport through the ocean surface, freshwater input would have to be about 50 times larger than the typical value in this region (4.5 m/month). This indicates that the surface freshwater input cannot contribute significantly to the imbalance.

Overall, these estimates lead to the conclusion that the processes discussed herein (i-iv) may not contribute significantly to the uncertainties in the transport budget.

## 3.3 Temporal Variability of the Benguela Current

The characteristics of the variability of the transports shown in Figure 5 are assessed with a wavelet based spectral analysis (Figure 6). The spectral energy of transports from the three sections covers a broad range of periods, mainly within 3 months to one year. The influence of the Agulhas rings are clearly visible with significant energies in 3 to 7 months range in certain years (Figure 6a,b).

In terms of seasonality, a strong annual cycle is visible at $30^{o}$S in 2006 to 2011 (Figure 6a,d). In most of the other years, the spectral density for the annual cycle still has a maximum of 3 to 4 $Sv^2$/cycle, but it does not reach the level of significance. In contrast to this, at $35^{o}$S the level of significance for an annual cycle is only reached in 1996 to 1998 (Figure 6b) and the energy in other years is mostly lower than at $30^{o}$S (Figure 6d).

The weakness of the annual cycle and the energy at mesoscale periods can be attributed to the impact of the Agulhas rings. About 5 to 6 Agulhas rings cross the Cape Basin region annually and they typically translate in a northwesterly direction. Overall, the energy at $35^{o}$S is slightly higher than at $30^{o}$S within this frequency band. The differences between these two latitudes are consistent with the fact that the Agulhas rings have a larger impact at $35^{o}$S than at $30^{o}$S, as can be seen in the Hovmöller diagrams (Figure 7). In addition to this, the Hovmöller diagrams reveal why the energy at mesoscale periods (3-7 months), as derived from the time series of the transport in the Benguela Current, is not always significant. Individual Agulhas rings only have an impact on a small part of the longitude range used in the transport computation. Therefore, their signal becomes relatively weak (Figure 7). Also, it has to be noted that the monthly Argo & SSH data set cannot resolve periods smaller than 2 months, therefore, transport time series may miss some of the high frequency variability due to these rings. Periods with significant spectral energy at mesoscale frequencies can be attributed to the presence of more than one Agulhas ring at a given latitude. An example for this can be seen in 1995 to 1996 at $35^{o}$S (Figure 7a).

For the zonal transport across line AC, Figure 6c reveals a dominance of frequencies at periods of 3 to 7 months that is more persistent than at $30^{o}$S and at $35^{o}$S. This is because the signal from the Agulhas rings can be captured more easily in this meridional section (AC) than the zonal sections at AB and AC. The meridional section AC is about three times shorter than the

zonal ones and its length is close to the typical diameter of the Agulhas rings. It is noted that the zonal transport does not exhibit

a statistically significant annual peak in any year. A major reason for this is, again, the prevalence of Agulhas rings. Figure 8c, which shows the ratio of eddy kinetic energy (Figure 8a) and mean kinetic energy (Figure 8b), reveals that the rings typically cross line AC. Herein, we used the assumption that powerful Agulhas rings are characterized by an eddy kinetic energy that is mostly at least 10 times larger than the mean kinetic energy. The exception of this is the region where the Benguela Current has the highest velocities (Figure 2).

**3.4    Wind Forcing and the Sverdrup Balance**

The Sverdrup relation gives a zero'th order understanding of the wind forcing and vertically integrated meridional transport in an open ocean. Validity of the Sverdrup relation has been analyzed in many studies both using observations (e.g. Gray and Riser (2014)) and model simulations (e.g. Thomas et al. (2014); Wunsch (2011)). The focus of these studies was mostly to determine whether the Sverdrup balance holds in the open ocean, but they did not focus on the eastern boundary

region. Nevertheless, Gray and Riser (2014) stated that the Sverdrup balance cannot explain the observed transport near the eastern boundary. Small et al. (2015), using regional ocean model, showed that an approximate Sverdrup balance holds close to the eastern boundary while it underestimates the transport in the region where the Benguela Current is strong. Using the Regional Ocean Modeling System (Shchepetkin and McWilliams, 2005) Veitch et al. (2009) also found that Benguela Current transport is larger than that derived from the Sverdrup balance (their Figure 2a,b).

While keeping in mind that the Sverdrup balance has low skill with respect to reproducing transports from observations or models, it is used herein to investigate what impact the wind field has on the variability of the transport. To accomplish this, the curl of the wind stress is calculated using monthly mean ERA-Interim wind fields for the same time period as the observations. Figure 9a shows the climatological mean of the curl of wind stress as well as the wind-stress vectors. The latter reveal, as

expected, that the direction of the winds in the region is similar to the direction of the transport (Figure 2b). The wind-driven transport can be calculated with the Sverdrup equation

$$M_y = |\frac{\nabla \times \tau}{\rho \beta}|,$$

where, $M_y$ is the Sverdrup transport, $\tau$ is the wind stress and $\beta = \frac{\partial f}{\partial y}$ is the meridional gradient of the Coriolis Parameter $f$, and $\rho$ is density.

Figure 9b shows the Sverdrup stream function as derived by integrating $M_y$ from the African coast to the west. It indicates that the transport across $30^o$S and $35^o$S is about 2 Sv which, as expected, is much smaller than the mean meridional transports obtained from Argo & SSH at these latitudes.

A challenge in this region is the inflow of water from the Indian ocean, typically about 14.5 Sv in the upper 1000 m (Richardson, 2007), via the energetic Agulhas rings and leakage. In addition, the Sverdrup balance may have a different depth for the wind forced layer than 800 m used for estimating the meridional transport. However, values from a study by Schmid et al. (2000), who estimated the depth of the gyre using a ventilated thermocline approach by Luyten and Stommel (1986), showed values of 700 - 800 m for the box ABCD.

To gain further insight into the wind-driven variability of the meridional transport, the Sverdrup transport across AB and CD is integrated zonally and its anomalies are compared to the anomalies of the observed transport after smoothing with a 18-month running mean filter and normalizing (Figure 10). Across AB (at $30^o$S) the time series exhibits about the same variability, however, this is not true at $35^o$S.

These results suggest that, for interannual variability, the wind field forces the circulation at $30^o$S with a two month lag. At $35^o$ this is not the case, most likely due to the large impact of the inflow from the Indian Ocean. A significant portion of the variability, however, remains unexplained, even at $30^o$, where the Sverdrup theory works relatively well. Therefore processes such as density variation, eddy shedding, and remote forcing remain important to the overall variability in the highly dynamic Cape Basin region.

## 4   Summary and Conclusions

The objective of this study is to increase the knowledge of the structure and variability of the flow in the eastern limb of the Atlantic Meridional Overturning Circulation in the subtropical South Atlantic by deriving and studying a 23 year long time series of transport estimates in the Benguela Current region from observations. Historically, the coverage with observations of this nature was relatively sparse in this region. The relatively long time series of velocities and integrated transports as well as the derived results will be valuable for validating models in this dynamic region. This study provides mean volume transports every $0.5^o$ between $35^o$S and $25^o$S and thus greatly expands the knowledge on the latitude dependence of the integrated meridional transport of the Benguela Current. In addition to that, it allows a more detailed analysis of its temporal variability than was possible previously.

The mean volume transports are mostly in good agreement with previous estimates when comparing those with similar integration limits. It is found that the meridional transports of the Benguela Current are relatively higher south of $31^o$S, and relatively lower values are typically seen north of $28^o$S where the zonal component of this current is stronger.

The mean meridional transport of the Benguela Current in the upper 800 m ranges from 9±3 Sv to 23±3 Sv (Figure 4; whenever possible, transports are represented as mean ± standard deviation). In the upper 1000 m, the range is 10±3 Sv to 26±3 Sv. The transports in the upper 1000 m are between 5% and 25% (1% and 5%) higher south (north) of $29^o$S than those

integrated over the upper 800 m. The agreement with estimates from previous studies is mostly good if one keeps in mind that most of them are for synoptic sections and use different vertical integration limits. In addition, their zonal integration limit in the west varies from Greenwich Meridian to $3^{o}$E. For example Clement and Gordon (1995) and Stramma and Peterson (1989) both use Greenwich Meridian as the western edge of the Benguela Current at $32^{o}$S, where as Garzoli and Gordon (1996) integrated to $3^{o}$E to obtain transport at $30^{o}$S.

A recirculation cell is observed in the climatologies of 15 m velocity and integrated transport in the upper 800 m between Walvis Ridge and Vema Seamount. The recirculation cell is centred at $6^{o}$E and $33^{o}$S and might be formed due to the interaction of eddies with the bathymetry in this region.

In terms of the variability, both $30^{o}$S and $35^{o}$S do not reveal a strong annual cycle. However, there are periods where the energy density reaches the level of significance at both latitudes. Overall, the energy density at $30^{o}$S has more energy in this frequency band than $35^{o}$S. With respect to mesoscale variability, the time series at $30^{o}$S and $35^{o}$S exhibit large energies in 3 to 7 months period due to the influence of the Agulhas rings. However, this energy does not reach the level of significance in many years because it is based on a zonal integral of the transport as explained in section 3.3.

At $30^{o}$S, the normalized anomalies of smoothed meridional transports from the observations and from the Sverdrup balance exhibit strong correlations (0.7), with the latter leading the former by two months. Smoothed time series of these two transports show similar interannual variability at $30^{o}$S. In contrast to this, variability of the observed transport across $35^{o}$S is significantly different than the Sverdrup balance. Therefore, this leads to the conclusion that the variability of the meridional transport at $30^{o}$S is significantly impacted by the local wind forcing while this is not the case at $35^{o}$S. Discrepancies at $35^{o}$S can be explained by the impact of the water flowing from the Indian Ocean into the South Atlantic.

To understand the flow pattern in the Cape Basin region transport budgets are estimated in two different boxes between $30^{o}$S and $35^{o}$S, and $3^{o}$E ($8^{o}$E) and the eastern boundary near the African Coast. It is found that the local topography near the Walvis Ridge and a recirculation cell centered at $33^{o}$S, $6^{o}$E can significantly influence the volume transport budget in the upper 800 m. Cyclonic and anticyclonic eddies can also cause large anomalies in the budget. However, away from the coast at $\sim$1000m isobath coastal upwelling events seem not to influence the transport budget.

Recent studies with climate models suggest an intensification of the westerlies in the Southern Ocean (Cai, 2006; Yang et al., 2016); and it has been shown that the increasing westerlies in the Southern Ocean result in an intensification of the Agulhas Current and its leakage since 1965 to about 1990 (Biastoch et al., 2009, 2015). Both Biastoch et al. (2015) and Beal and Elipot (2016) did not find a positive trend in the Agulhas transport from 1993 on. Consistent with this, the time series presented herein do not indicate that there is a trend in the transport of the Benguela Current or the transport across $3^{o}$E between $30^{o}$S and $35^{o}$S.

*Data availability.* An earlier version of this data set is available at $http://www.aoml.noaa.gov/phod/samoc_argo_altimetry/index.php$; the new extended dataset is aviable upon request.

*Competing interests.* none

*Acknowledgements.* This paper was funded by the Climate Observation Division, Climate Program Office, Climate Monitoring Program,

National Oceanic and Atmospheric Administration, U.S. Department of Commerce and the Atlantic Oceanographic and Meteorological Laboratory of the National Oceanic and Atmospheric Administration. This research was also carried out in part under the auspices of the Co-operative Institute for Marine and Atmospheric Studies (CIMAS), a cooperative institute of the University of Miami and the National Oceanic and Atmospheric Administration (NOAA), cooperative agreement NA'0OAR432013. The authors would like to thank the researchers and technicians involved in the Argo project for their contributions to generating a high-quality global sub-surface data set. Argo data were ob-

tained from the Global Data Assembly Centre (Argo GDAC,http://dx.doi.org/10.12770/71b7b0ed-1e3a-4ebc-8e3b-b5b363112f2a. Altimeter products were produced by Ssalto/Duacs and distributed by AVISO, with support from *Cnes* (http://www.aviso.altimetry.fr/ducas/) and the 3-D velocity data can be obtained from the PHOD website.

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

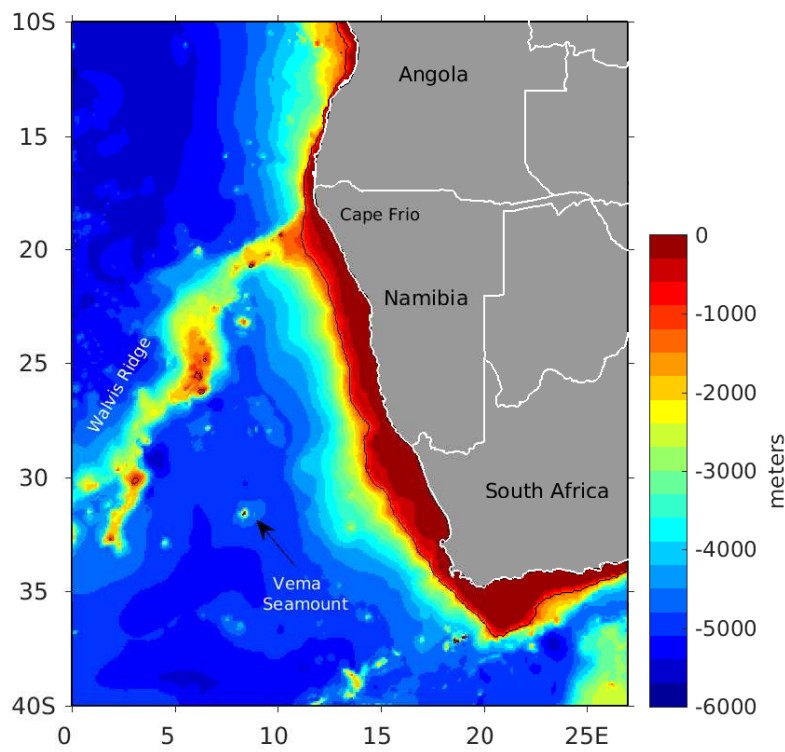

**Figure 1.** A map of the Benguela Current region showing local topography and boundaries.

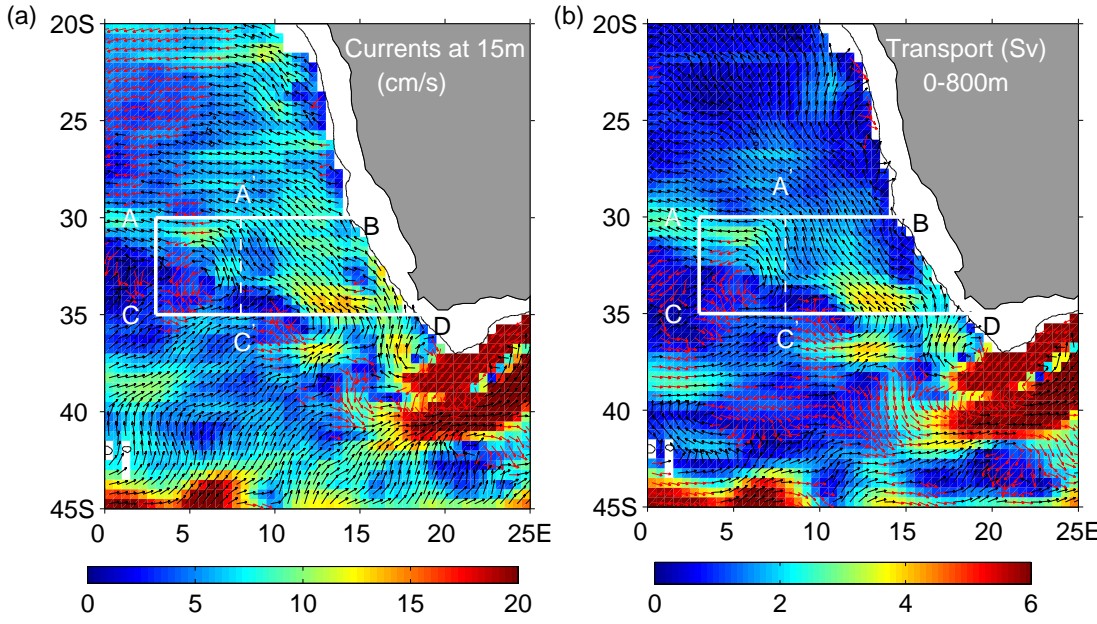

**Figure 2.** Climatologies of the flow at 15 m (a) and transports (b) in the upper 800 m from Argo & SSH. Red and black arrows indicate flow to the north and the south, respectively, and the shading represents magnitude. A transport budget is derived for two regions: ABCD (extending from $3^oE$ to the 1000 m isobath near the eastern boundary) and A'BC'D (extending from $8^oE$ to the 1000 m isobath near the eastern boundary).

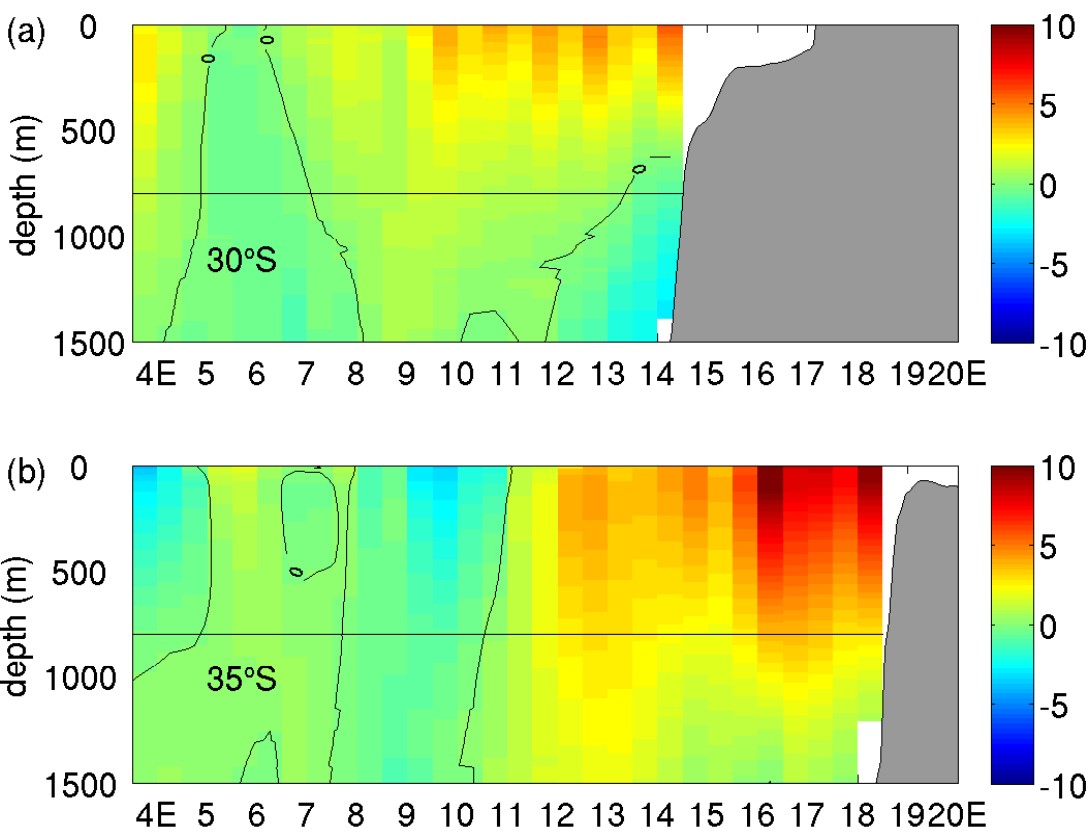

**Figure 3.** (a) Climatological mean of the meridional geostrophic velocity across $30^{o}$S (a) and $35^{o}$S (b). Black lines are the contours of zero velocity and the black straight line marks the depth of 800 m.

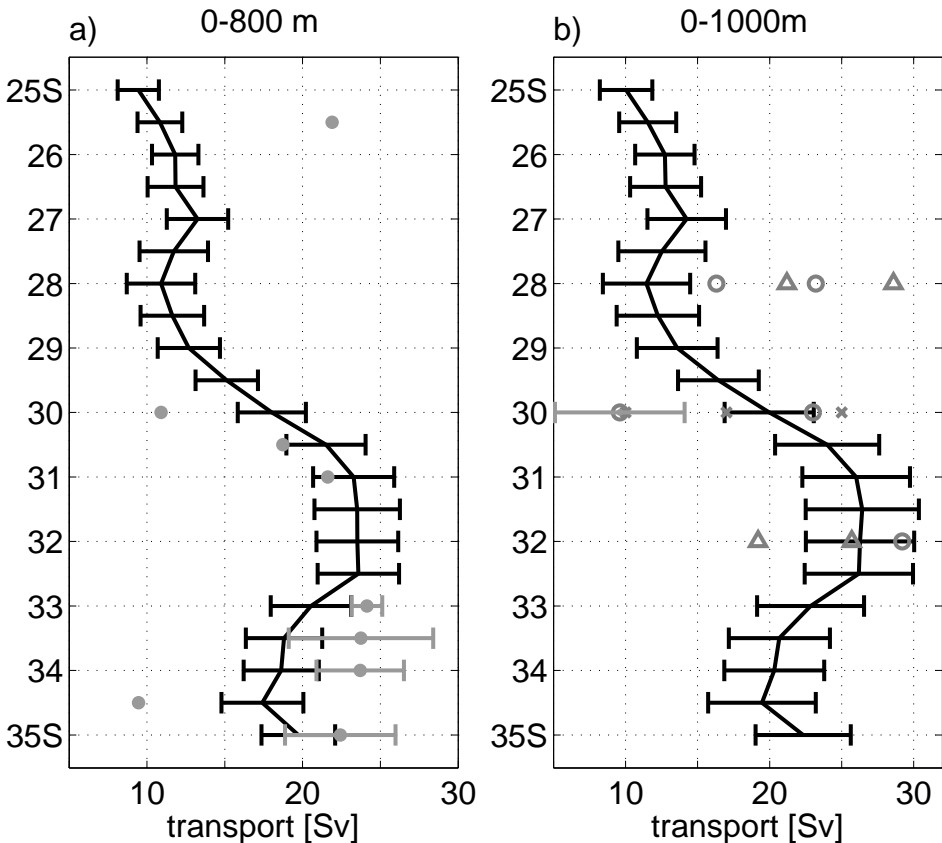

**Figure 4.** Latitude dependence of climatological mean of the meridional transport in the Benguela Current (between $3^o$E and the eastern boundary) in the upper 800 m (a) and 1000 m (b). Gray dots are based on velocity transects derived by Garzoli et al. (2013) for the purpose of estimating the MOC transports in the South Atlantic. Other gray symbols are based on transport estimates from three studies: Garzoli and Gordon (1996), Clement and Gordon (1995), and Stramma and Peterson (1989). Gray triangles (circles) indicate that the western integration limit was the Greenwich Meridian (the western edge of the Benguela Current). All other gray symbols represent estimates based on a western integration limit at $3^o$E. Gray error bars are shown where an estimate was derived from multiple transects or a time series. The vertical integration limits for Stramma and Peterson (1989) (shown in (b)) range from 940 m to 1180 m, with the largest values used at $32^o$S and the smaller ones in $23^o$S to $24^o$S, and they derived each transport for two different reference levels as well as two different western integration limits. All error bars represent standard deviations.

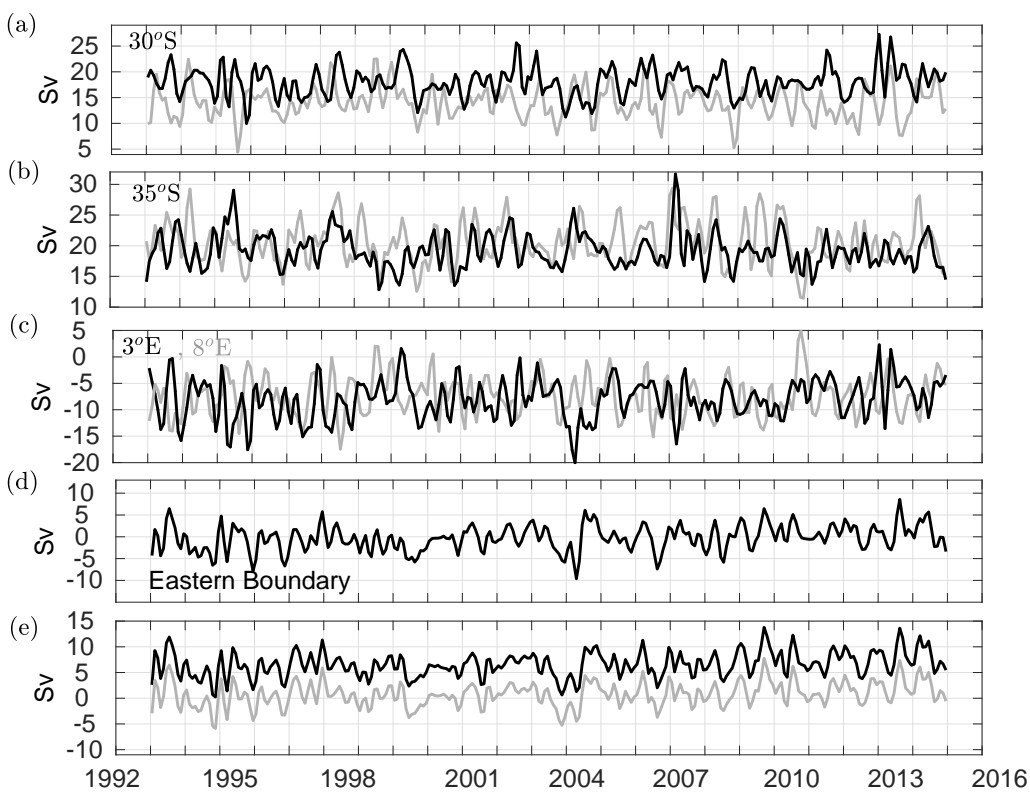

**Figure 5.** Time series of transports in the upper 800 m. (a) meridional transport of the Benguela Current at $30^o$S (black - across line AB; gray across line A'B, see Figure 2). (b) meridional transport of the Benguela Current at $35^o$S (across line CD in black; across line C'D in gray). (c) zonal transport at $3^o$E (across line AC; across line A'C'). (d) Cross-shelf transport at the eastern boundary near the African coast across 1000 m isobath (across line BD). (e) Transport imbalance for ABCD (black) and A'BC'D (gray).

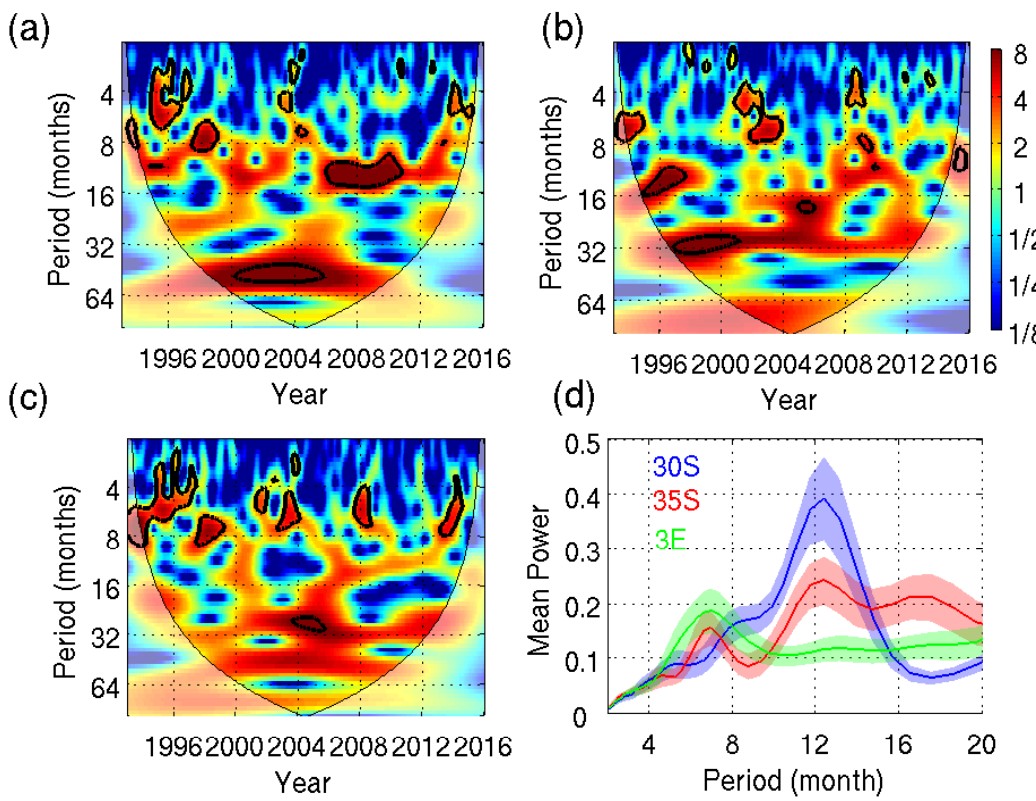

**Figure 6.** Wavelet spectral density of meridional transports at (a) $30^o$S and (b) $35^o$S, and (c) zonal transport at $3^o$E. (d) Mean wavelet power spectra at $30^o$S (blue), $35^o$S (red) and $3^o$E (green). The black contours are the 95% confidence interval. The values outside the cone of influence (blurred colors) indicate where the edge errors dominate.

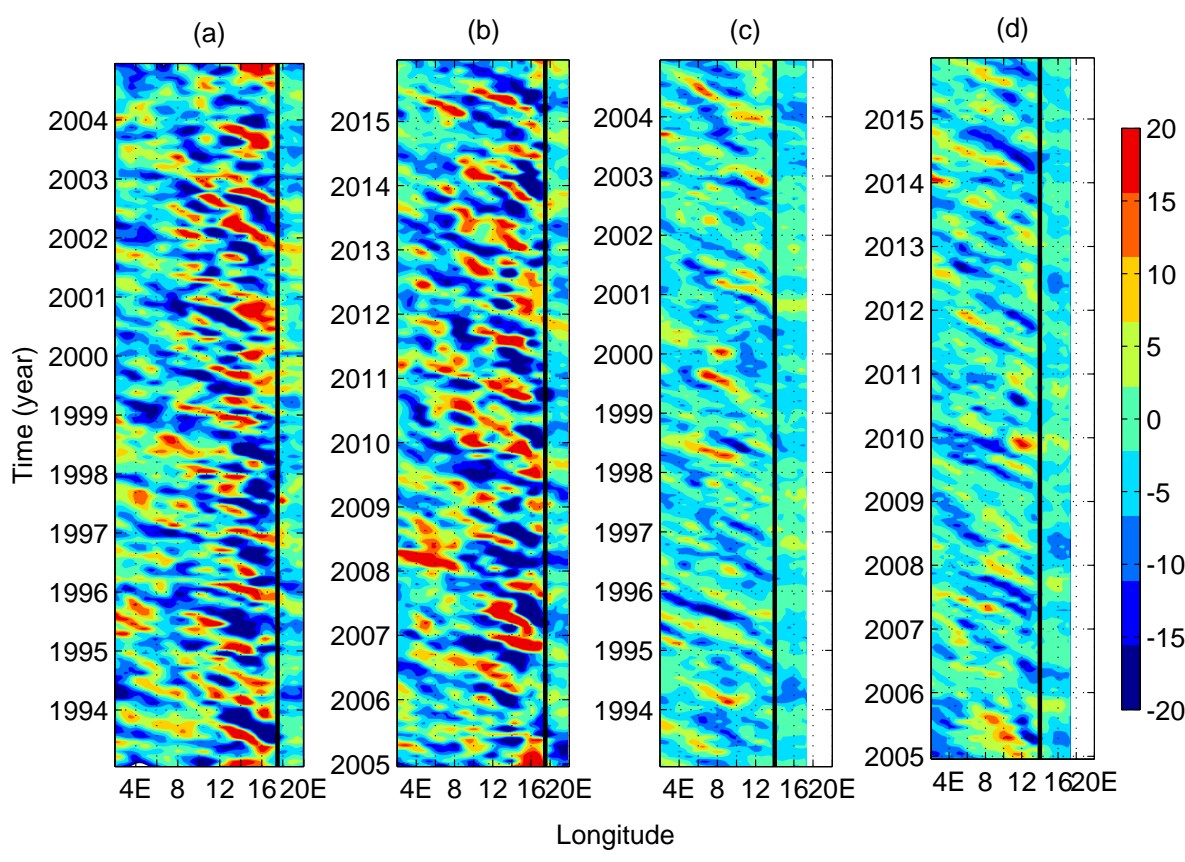

**Figure 7.** Hovmöller diagram of monthly sea surface height anomalies (in cm) across $35^{o}$S (a,b) and $30^{o}$S (c,d). Anomalies are stronger west of the black line located at $17.5^{o}$E for $35^{o}$S and at $14^{o}$E for $30^{o}$S.

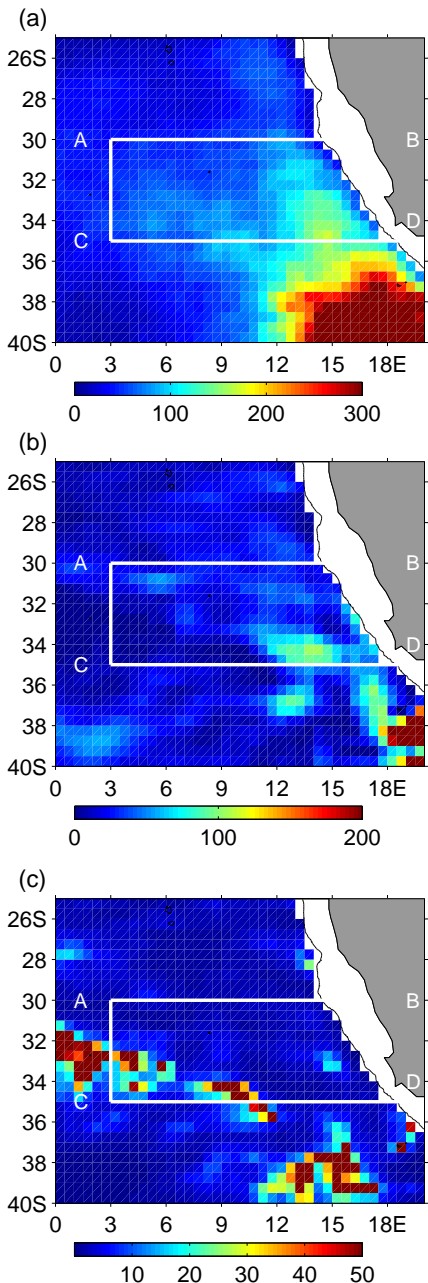

**Figure 8.** Climatologies of (a) eddy kinetic energy in $(cm/s)^2$ , (b) mean kinetic energy in $(cm/s)^2$, and (c) the ratio of eddy and mean kinetic energy.

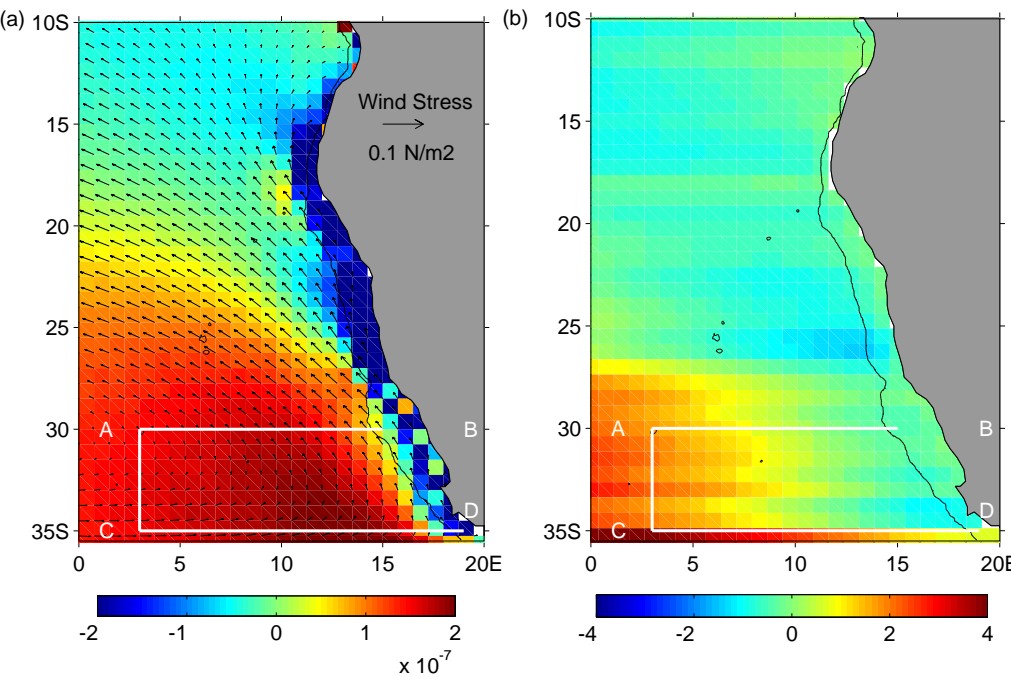

**Figure 9.** (a) Climatologies of ERA interim wind stress curl (in color) and the wind stress (arrows). (b) Climatology of the Sverdrup stream function (color).

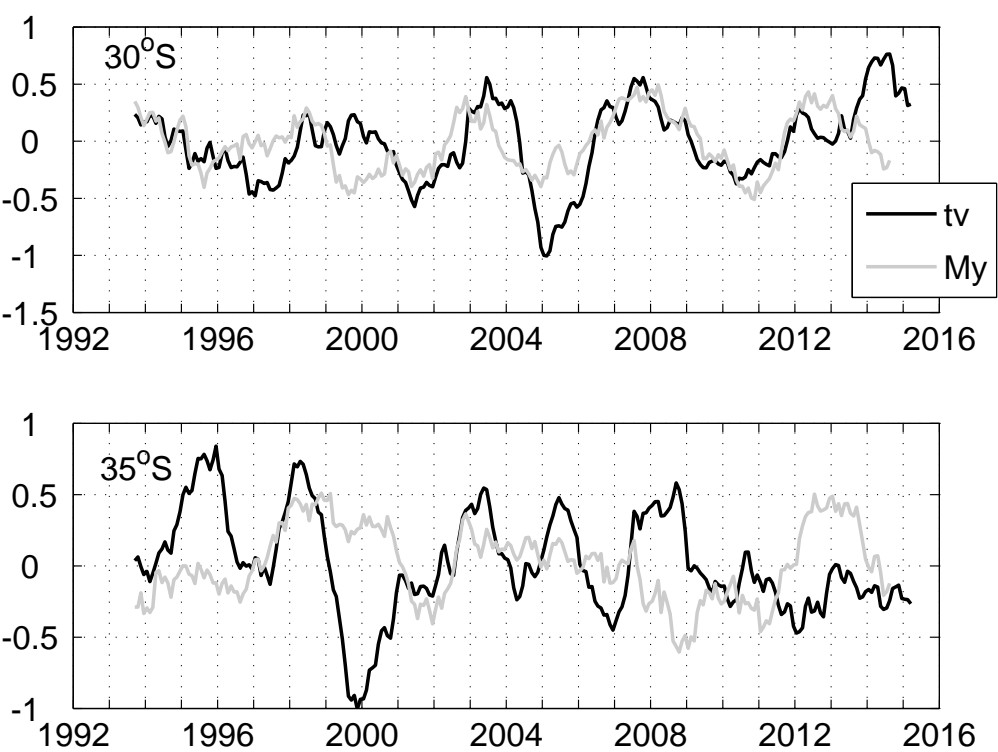

**Figure 10.** Zonally integrated normalized anomalies $M_y$ (gray) and meridional transport (tv, black) at $30^o$S and $35^o$S. Both the time series are smoothed using a 18-month running mean filter.