# Peer review of "A Study of the Variability of Volume Transport of the Benguela Current"

_Ocean Science, 2017_

## Referee Comment (RC1) · Anonymous Referee #1 · 8 Sep 2017

Referee comment on **A study of the variability of the Benguela Current**, by Sudip Majunder and Claudia Schmid.

**General comment**

Driven mainly by economic interests, the countries along the west coast of southern Africa have, for many decades, conducted dedicated marine research efforts to survey and understand aspects of the Benguela Current System. Much of the research was spatially and temporally constrained by infrastructure (e.g. research vessels) and funding, and the only long-term data sets that became available included SST observations, tidal records and fisheries/biological parameters.

Ocean topography, ARGO floats and NCEP data is now deployed by the authors to provide a whole new insight ("bigger picture") into the three-dimensional flow and variability of the area. Their approach is therefore novel, and the analysis and presentation are good. The results are therefore considered relevant.

**Suggestions and basic comments**

a. Amendment of title

Much is known about other (non-physical) aspects of the Benguela Current. It is suggested that the title should indicate that the manuscript is confined to the *physical* aspects (current, flow, transport), and maybe even add the years covered.

b. Suggested links with existing knowledge

The title of the manuscript focuses on the *physical* aspects of the Benguela's variability (currents, volume transport). Considering the strong link that exists between the physics and the fisheries, biochemical and environmental issues of the region, and to indicate that interested readers can follow up on the extensive research that has been conducted and reported in those fields, the Addendum below provides some information.

**Definition of the Benguela and regional features**

- It would be of great use to the reader to know how the authors delineate the Benguela in terms of relevant environmental parameters, such as speed, volume transport, flow direction, or latitude/longitude.
- It would also help if some of the geographic features, such as Walvis Ridge, are identified on a chart of the area. A map of the area [such as the one by Shannon V (2006) *A plan comes together*. in: Benguela, Predicting a Large Marine Ecosystem. Large Marine Ecosystems Vol 14, Elsevier BV, p 4] would be very useful for readers not familiar with the local topography and nomenclature.
- p 3 line 2: The reader is uncertain why names like *Cape Frio* and *Cape Agulhas* are mentioned (instead of just their latitudes) without identifying them on a map of the area.
- p 5 line 22 and 23: It seems that the distinction between "steady" and "transient" flow is based on the spatial variability of the flow direction (?). In the absence of a reasonably clear delineation of the Benguela Current there is some confusion whether the "steady flow" is synonymous with the Benguela Current.

- p 5 line 30: "...may occur in two branches." The 15 m flow pattern (Figure 1a) suggests that there is a third contributory branch crossing 35°S northeastward at about 7°E. This seems less distinct in Figure 1b, so it may be largely a surface feature.

**Definitions and clarity of terminology**

- The authors tend to use "northward" (e.g. p 2) when the flow is largely northwestward. E.g. is "meridional" the northward component of the NW flow? This confusion between the flow and the flow components must be avoided.
- p 4 line 3: readers may know what Sv means, but as a non-SI unit and for the sake of completeness it should be defined when first used.
- p 6 line 3-6: The vertical sections in Figure 2 display only the meridional component of the flow. Low values do not necessarily mean that the flow as a whole is low, but just that the meridional component is low. It can therefore not be used to differentiate between the "steady" flow and the "transient" flow (i.e the flow with a stronger zonal flow component need not necessarily be less steady).

**Clarity of approach and presentation**

- p 6 line 15-18: It is understood that there is data gap in the inshore, shallower regions. However, the authors can only extrapolate the flow eastward into areas shallower than 1000m  (in terms of speed and direction) if there is a justifiable basis (e.g. references).
- p 6 line 17-18: Using climatological averages a meridional flow of 1.8-2.0 Sv is derived for the inshore region at the latitudes of 30° and 35° S. So, if there is approximately the same volume of water entering at 35°S than is leaving at 30°S, how is this related to the value of 7 Sv leaving the coastal area westwards (reported in p 7 line 3)?
- p 6 line 20 and further: Considering that the Benguela Current direction is around NW, the volume transports computed along the latitudinal lines at 30°S and 35°S reflect only about 70% of the whole transport. The authors should indicate why lines oriented in a SW-NE direction were not chosen, and the effect on the computed transports. Has the orientation of the lines not perpendicular to the main flow axis of the Benguela been taken into account when comparing the transport results with other authors (line 25-27)?
- p 7 line 3: Flow in shelf area: Figure 1 indicates the location of  points B and D, but why are they so far inland? Where exactly is "parallel to the shelf break", or do the authors mean "at the shelf break"?
    - Were the flows reported across AC and BD corrected for their mutual difference in orientation?
    - Visual inspection of Fig 1a and b suggests that the vectors along the eastern perimeter (closest to the shelf edge) are oriented largely parallel to the shelf edge, except in 30/31° and 34/35° where flow is oriented slightly onto the shelf, and at 31/32° where the flow is oriented with an off-shelf component.
    - The net westward (?) flow of 7 Sv  at BD therefore seems to be largely associated with the off-shelf flow at 31/32°. The chart of Shannon (mentioned above) indicates a south-eastward flow on the shelf at 30°, creating the possibility of confluence  of southward and northward flows and a solution to the transport imbalance.
- p 7 line 5: Shorter-period anomalies: The computation of the climatic volume balance is a huge step forward in the insight of the flow/transport of the area and the Benguela Current (Fig. 4). The authors admit that, at times, the budget is

unbalanced. There seem to be events that not only imbalance the climatology, but are in themselves huge anomalies (of the order of the average condition). The following are examples:

- o In 2005 there occurred a westward flow of approximately 20 Sv across AC. This was the largest flow of the data record, and seemed to coincide with an equally large westward flow across the eastern boundary. These flows were 2-3 times the average. While the flow across AB was slightly below average, the flow across CD was about 5 Sv larger than average.
- o In 2014 two large flow pulses northward at AB coincided with similar eastward events across AC.

The authors indicated some possible reasons for transport discrepancies. If the events mentioned above occurred more-or-less simultaneously (as they appear) a specific investigation to pin down what happened, is called for.

- p 11 line 12: Impact of local wind forcing: Lutjeharms and Meeuwis (1987, *S Afr J mar Sci* **5**, 51-62) indicated that the coastal area between 25° and 30°S seems to have the highest, upwelling-favourable wind speed of all locations along the southwestern coast of Africa.There is a significant decrease in the vicinity of 34°S.
- p 20: The order of (a),...(d) at the top of the figure need checking?

**Issues of expression, grammar, etc**

- p 2 line 5: "..long-term velocity observations" [there are other fisheries-biological data records and satellite SST that continue for many years/decades]
- p 2 line 9: insert of after study
- p 3 line 16: "... two thirds ... contribute..."
- p 4 line 8: "Further east ..." Shouldn't this be west?
- p 4 line 9: conducted
- p 4 line 17: "..no long-term measurements of this current's flow/transport/dynamics.. are available"
- p 4 line 33: "0.5x0.5 grid": It is suggested to change this to 0.5°x0.5° grid"
- p 5 line 8: "..dynamic height does not change.."
- p 5 line 15: insert "and" after coast
- p 5 line 26: The convention is that positions should be reported as (lat, long) and, not as (long, lat).
- p 5 line 32. I don't see any "zonal velocities" in die area indicated by the authors. Do the authors mean "zonal velocity components" when they say "zonal velocity"? This was also mentioned above, and it is suggested that the authors rigorously identify and reword such misnomers.
- p 6 line 9. It is possible that Figure 2b shows a weak signal around 7°E corresponding to the third tributary referred to before (p 5 line 30).
- p 6 line 11-12: So the westward flow at 30°S is still the Benguela Current?
- p 6 line 12: insert "in" after "resulting"
- p 6 line 20: The authors should indicate the origin of the standard deviation.
- p 6 line 32: transports
- p 8 last paragraph: I liked the results concerning the course of Agulhas rings.
- p 9 line 4: cannot
- p 10 and 11: The Summary and conclusions are in order
- p 11 line 2: Vema
- p16: "...the black straight line marks the depth.."

**Addendum**

Without trying to mention all the research papers (especially the most recent ones), I recall that there were 5 compendia (below), based on symposia, where some of the results have been summarised. I draw attention (as examples only) to three papers with their full reference, contained in some of the books:

- *The Benguela and comparable ecosystems* (ed. Payne, Gulland and Brink),1987, 956pp
- *Benguela trophic functioning* (ed. Payne, Brink, Mann and Hilborn), 1992, 1108 pp. [Shillington FA, L Hutchings, TA Probyn, HN Waldron and WT Peterson (1992) Filaments and the Benguela Frontal zone: Offshore advection or recirculation loops? *S. Afr. J. mar. Sci*. **12**, 207-218.
- *Benguela dynamics. Impacts of variability on shelf environments and their living resources*. (ed. Pillar, Moloney, Payne and Shillington), 1998, 512 pp. [Strub PT, FA Shillington, C James and S Weeks (1998) Satellite comparison of the seasonal circulation in the Benguela and California current systems. *S. Afr. J. mar. Sci*. **19**, 99-112]
- *Ecosystem approaches to fisheries in the Southern Benguela* (ed. Shannon, Cochrane and Pillar), 2004, 328 pp.
- *Benguela: Protecting a large Marine Ecosystem* (ed. Shannon, Hempel, Malanotte-Rizzoli, Moloney and Woods), 2006, Elsevier B V, 410 pp. [Shillington FA, CJC Reason, M Duncombe Rae, P Florenchie and P Penven (2006) Large scale physical variability of the Benguela Current Large Marine Ecosystem (BCLME). p 49-70]

If required, contact may also be taken up with the University of Cape Town and (especially) the Coastal and Oceans Branch of the Department of Environmental Affairs in Cape Town. This Branch and its predecessor *Sea Fisheries Research Institute* has been responsible for the bulk of the multi-disciplinary research in the Benguela region.

---

## Referee Comment (RC2) · J. Kämpf (Referee) · 12 Sep 2017

Given the excellent review of anonymous reviewer #1, I only have few points to add. While the authors seem to have derived an excellent data set of the horizontal velocity field in the upper 800 m of the water column (only in regions >1000 m deep!!!), the analysis of the transport variability has some loose ends that need to be addressed. The required revisions may be classified as "major".

Major points:

1) Which process creates significant negative (westward) transports of 7 +/- 4 Sv across the B-D transect (perpendicular to the shelf break)? There a peaks exceeding 15 Sv in magnitude! Where does this water volume come from? Is this a downstream

signature of the upwelling jet known as "Columbine Jet"? Interestingly, there seem to be instances of high correlation between the zonal transport perpendicular to the shelf break and that at 3W. Could some of these variations be caused by coastal wind variations (driving the upwelling jet)? In my view this is an important feature that needs to be studied and explained as part of this paper.

2) Total volume transport is only one limited aspect of scientific interest. Heat and freshwater transports and also nutrient fluxes are probably of similar or even higher significance. Changes in the baroclinic flow structure would be another point worth exploring. Have the authors considered to extend their analysis to those features? If not, I suggest the paper be renamed to "A Study of the Variability of Total Volume Transports of the Benguela Current." As it is, the current title may be misleading.

3) Given that the authors have developed a complete data set, I am confused as to why the authors used four different southern latitudes of 28, 30, 31 and 35. Why not only 28 and 31, or only 30 and 35? Why not a continuous section at 1 degree steps?

4) The authors' reference to "transport in the upper 800 m" can be misleading. The fact that the analysis excludes regions <1000 m in depth needs to be stated in the abstract and elsewhere.

Other points:

Page 9 => found that Benguela Current transport is larger than "that derived from the" Sverdrup balance (their Figure 2a,b). => Insert suggested phrase.

Sverdrup Gyre => Where does this terminology come from? Technically this term is incorrect because the Sverdrup balance only describes most but not all of the dynamics inherent with subtropical gyres. You wouldn't call them "Stommel Gyres" either, would you? One option is to rename this to "Sverdrup balance", but perhaps a better option would be to use "Ekman pumping" as a parameter in the analysis, which would avoid the unnecessary discussion about the validity of the Sverdrup balance in the region.

---

## Author Comment (AC2) · 24 Nov 2017

**Response to reviewer 2:**

Thank you for the very helpful comments and suggestions.

Major points:
*1) Which process creates significant negative (westward) transports of 7 +/- 4 Sv across the B-D transect (perpendicular to the shelf break)? There a peaks exceeding 15 Sv in magnitude! Where does this water volume come from? Is this a downstream signature of the upwelling jet known as "Columbine Jet"? Interestingly, there seem to be instances of high correlation between the zonal transport perpendicular to the shelf break and that at 3W. Could some of these variations be caused by coastal wind variations (driving the upwelling jet)? In my view this is an important feature that needs to be studied and explained as part of this paper.*

Thank you for pointing this out. Based on the suggestions from reviewer 1 we have updated our analysis on the transport budget and discuss several events. In the previous version the slope of transect BD was not handled correctly. Now, the cross-shelf transport is estimated for line BD.
This transport is smaller than the previous one, with a southwestward mean of about 0.3 Sv. However, some large transports occur (<8 Sv). It is found that these anomalies can mostly be attributed to cyclonic and anticyclonic eddies. The revised text can be found in lines p8, lines 4-19.

*2) Total volume transport is only one limited aspect of scientific interest. Heat and freshwater transports and also nutrient fluxes are probably of similar or even higher significance. Changes in the baroclinic flow structure would be another point worth exploring. Have the authors considered to extend their analysis to those features? If not, I suggest the paper be renamed to "A Study of the Variability of Total Volume Transports of the Benguela Current." As it is, the current title may be misleading.*

Thank you for the suggestion. We have changed the title of the manuscript to : A Study of the Variability Volume Transport of the Benguela Current.

*3) Given that the authors have developed a complete data set, I am confused as to why the authors used four different southern latitudes of 28, 30, 31 and 35. Why not only 28 and 31, or only 30 and 35? Why not a continuous section at 1 degree steps?*

Mean values of transport are shown for a continuous section at 1 degree steps in figure 4.
For the transport budget we chose an area enclosed between 30S and 35S, and 3E and 1000m isobath parallel to the African coast. This area is very dynamic since both Indian Ocean water and the water from the south Atlantic Current interact here; also, Agulhas eddies use this region as a corridor to the northwest. Therefore in order to explore the limitation/capabilities of our data set this region was chosen.

*4) The authors' reference to "transport in the upper 800 m" can be misleading. The fact that the analysis excludes regions <1000 m in depth needs to be stated in the abstract and elsewhere.*

Thank you for the suggestion, the manuscript has been revised. (See lines p2, line 6; p6, lines 23-25)

*Other points:*
*Page 9 => found that Benguela Current transport is larger than "that derived from the" Sverdrup balance (their Figure 2a,b). => Insert suggested phrase.*

Thank you for pointing this out. This is now corrected in the manuscript (p10, line 19).

*Sverdrup Gyre => Where does this terminology come from? Technically this term is incorrect because the Sverdrup balance only describes most but not all of the dynamics inherent with subtropical gyres. You wouldn't call them "Stommel Gyres" either, would you? One option is to rename this to "Sverdrup balance", but perhaps a better option would be to use "Ekman pumping" as a parameter in the analysis, which would avoid the unnecessary discussion about the validity of the Sverdrup balance in the region.*

Thank you for the clarification, the manuscript has been updated on this. ( section title 3.4, p11 line3)

---

## Author Comment (AC1)

**Response to reviewer 1:**

Thank you for the very helpful comments and suggestions.

**Suggestions and basic comments**

**a. Amendment of the title**

Much is known about other (non-physical) aspects of the Benguela Current. It is suggested that the title should indicate that the manuscript is confined to the physical aspects (current, flow, transport), and maybe even add the years covered.

Thank you for pointing this out. We have now changed the title to:

"A Study of the Variability of Volume Transport of the Benguela Current"

**b. Suggested links with existing knowledge**

The title of the manuscript focuses on the physical aspects of the Benguela's variability (currents, volume transport). Considering the strong link that exists between the physics and the fisheries, biochemical and environmental issues of the region, and to indicate that interested readers can follow up on the extensive research that has been conducted and reported in those fields, the Addendum below provides some information.

**Definition of the Benguela and regional features**

• It would be of great use to the reader to know how the authors delineate the Benguela in terms of relevant environmental parameters, such as speed, volume transport, flow direction, or latitude/longitude.

The Benguela Current is defined by a longitude as the western edge of the current. Information about this has been added (first paragraph of section 3.1 on page 5).

- It would also help if some of the geographic features, such as Walvis Ridge, are identified on a chart of the area. A map of the area [such as the one by Shannon V (2006) A plan comes together. in: Benguela, Predicting a Large Marine Ecosystem. Large Marine Ecosystems Vol 14, Elsevier BV, p 4] would be very useful for readers not familiar with the local topography and nomenclature.
- *p* 3 line 2: The reader is uncertain why names like Cape Frio and Cape Agulhas are mentioned (instead of just their latitudes) without identifying them on a map of the area.

A map of the study area with names geographic features was added to the manuscript (figure 1).

• p 5 line 22 and 23: It seems that the distinction between "steady" and "transient" flow is based on the spatial variability of the flow direction (?). In the absence of a reasonably clear delineation of the Benguela Current there is some confusion whether the "steady flow" is synonymous with the Benguela Current.

Sorry for the confusion. We are now longer using the terms 'steady' and 'transient' in the manuscript. In the revised version these regions in the eastern and the western sides are now characterized in terms of

mean speed/kinetic energy. (p5, lines 28-32).

• *p* 5 line 30: "...may occur in two branches." The 15 m flow pattern (Figure 1a) suggests that there is a third contributory branch crossing 35°S northeastward at about 7°E. This seems less distinct in Figure 1b, so it may be largely a surface feature.

Thank you for pointing this out. This is now added in the manuscript. The feature at 7E is part of a meander of the South Atlantic Current. This feature does not contribute significantly to the Benguela Current. This can be seen in the climatological meridional velocity section at 35S in Figure 3b. (p6, lines 7-10)

**Definitions and clarity of terminology**

• The authors tend to use "northward" (e.g. p 2) when the flow is largely northwestward. E.g. is "meridional" the northward component of the NW flow? This confusion between the flow and the flow components must be avoided.

This is now corrected in the manuscript. (p2, line2; p1, line1)

• *p* 4 line 3: readers may know what Sv means, but as a non-SI unit and for the sake of completeness it should be defined when first used.

Agreed, now changed in the manuscript. (p4, line 5)

• *p* 6 line 3-6: The vertical sections in Figure 2 display only the meridional component of the flow. Low values do not necessarily mean that the flow as a whole is low, but just that the meridional component is low. It can therefore not be used to differentiate between the "steady" flow and the "transient" flow (i.e the flow with a stronger zonal flow component need not necessarily be less steady).

Thank you, this paragraph is now rephrased. (p6, lines 11-14)

**Clarity of approach and presentation**

• p 6 line 15-18: It is understood that there is data gap in the inshore, shallower regions. However, the authors can only extrapolate the flow eastward into areas shallower than 1000m (in terms of speed and direction) if there is a justifiable basis (e.g. references).

Thank you for pointing this out. We changed the text to clarify that the intent is to get a rough estimate of the uncertainty. By assuming that the velocity throughout the area without data is the same as at the easternmost observation we have, it is likely that we will overestimate this uncertainty. The reason for that is that the flow on the shelf is not always going to head in the same direction as the flow at the eastern end of the section.

• p 6 line 17-18: Using climatological averages a meridional flow of 1.8-2.0 Sv is derived for the inshore region at the latitudes of 30° and 35° S. So, if there is approximately the same volume of water entering at 35°S than is leaving at 30°S, how is this related to the value of 7 Sv leaving the coastal area westwards (reported in p 7 line 3)?

There was a problem with the method we used to derive the transport across line BD. We recalculated this transport and the budget (as described below). The newly estimated cross-shelf transport is found to be small about  $\sim$ 0.3+/-3Sv.

• p 6 line 20 and further: Considering that the Benguela Current direction is around NW, the volume transports computed along the latitudinal lines at 30°S and 35°S reflect only about 70% of the whole transport. The authors should indicate why lines oriented in a SW-NE direction were not chosen, and the effect on the computed transports. Has the orientation of the lines not perpendicular to the main flow axis of the Benguela been taken into account when comparing the transport results with other authors (line 25-27)?

For simplicity let us assume a steady flow V in the NW direction as depicted in the figure. The meridional component of the flow is  $V\cos\theta$  (in this case  $\theta$  is 45 degree to be precise). If we consider the depth of the water column to be h, and the flow is uniform in this depth range and along BC, the total transport in the NW direction is : V\*BC\*h= V\*ABcos $\theta$  \*h= transport in the meridional direction.

Therefore total transport of a north-westward flow is the same as the meridional transport.

• *p* 7 line 3: Flow in shelf area: Figure 1 indicates the location of points B and D, but why are they so far inland? Where exactly is "parallel to the shelf break", or do the authors mean "at the shelf break"?

B and D are now shifted to the eastern boundary of the box at 1000m isobath, to be precise. (p7, line 13, Figure 2)

• Were the flows reported across AC and BD corrected for their mutual difference in

**orientation?**

Thank you for pointing this out. The flow wasn't corrected. We obtained transport across BD by adding zonal transport of the eastern most grids between B and D in the upper 800m.

An estimate of the transport considering the slope of the African coast (cross-shelf transport) is now added in the manuscript. It is found that this new cross-shelf transport is relatively small (~0.3 Sv) compared to the transport (7 Sv) presented earlier. This relatively small cross-shelf transport is consistent with mean flow (Figure 2) pattern in this area; which is parallel to the coast line.

The cross-shelf transport is poorly correlated (0.15) with the along shore wind stress in this area. The cross-shelf transport modulates greatly by the passage of eddies from the Indian Ocean.

The decrease in the transport from the eastern boundary, however, results in a transport imbalance of ~6 Sv. To investigate this issue, we evaluated the transport budget for different areas and for different depths. It is found that an eddy (recirculation feature) at the western part of the box (Figure 2) and the topography near the Walvis Ridge are the main reason of this imbalance. To avoid the effect of these in the transport budget, we, therefore, choose an area between 8E and the 1000-isobath parallel to the African coast; the northern and the southern sides of the box remains the same. On an average the budget shows a balance within the error bars. However, some instances of imbalances are also identified. It is found that these imbalances are mostly caused by cyclonic and anticyclonic eddies.

- Visual inspection of Fig 1a and b suggests that the vectors along the eastern perimeter (closest to the shelf edge) are oriented largely parallel to the shelf edge, except in 30/31° and 34/35° where flow is oriented slightly onto the shelf, and at 31/32° where the flow is oriented with an off-shelf component.
- The net westward (?) flow of 7 Sv at BD therefore seems to be largely associated with the off-shelf flow at 31/32°. The chart of Shannon (mentioned above) indicates a south-eastward flow on the shelf at 30°, creating the possibility of confluence of southward and northward flows and a solution to the transport imbalance.

Based on the reviewer's suggestion we have computed cross-shelf transport along BD. With this new estimate we have a relatively stable budget and we do not see some of the anomalies that were noticeable in the earlier version of the manuscript. In the revised manuscript we describe few such anomalies in the budget. (p8, lines 4-17 and figure 5e)

- *p* 7 line 5: Shorter-period anomalies: The computation of the climatic volume balance is a huge step forward in the insight of the flow/transport of the area and the Benguela Current (Fig. 4). The authors admit that, at times, the budget is unbalanced. There seem to be events that not only imbalance the climatology, but are in themselves huge anomalies (of the order of the average condition). The following are examples:
  - In 2005 there occurred a westward flow of approximately 20 Sv across AC. This was the largest flow of the data record, and seemed to coincide with an equally large westward flow across the eastern boundary. These flows were 2-3 times the average. While the flow across AB was slightly below average, the flow across CD was about 5 Sv larger than average.

- In 2014 two large flow pulses northward at AB coincided with similar eastward events across AC.
- The authors indicated some possible reasons for transport discrepancies. If the events mentioned above occurred more-or-less simultaneously (as they appear) a specific investigation to pin down what happened, is called for.

With the new cross-shelf estimate the anomalies seen in the previous version of the manuscript are not prominent.

Based on the suggestions we have analyzed few events when the anomalies in the transport budget are relatively high. The highly variable cross-shelf transport (BD) has a mean of 0.3Sv and a standard deviation of 3Sv. In the updated manuscript, we describe few such events that show anomalies >6 Sv, during July 1993, April 2005, and August 2007.

It is found that both cyclonic and anticyclonic eddies intersect the southwest corner of the box and induce large northeastward/southwestward velocities; as a result the cross-shelf transport changes significantly. These eddies are very prominent in the Hovmoller diagram (Figure 7).

Along-shore wind does not seem to influence the cross-shelf transport in the region where the transport budget is computed. (p8, lines 4-17 and figure 5e)

• p 11 line 12: Impact of local wind forcing: Lutjeharms and Meeuwis (1987, S Afr J mar Sci 5, 51-62) indicated that the coastal area between 25° and 30°S seems to have the highest, upwelling-favourable wind speed of all locations along the southwestern coast of Africa. There is a significant decrease in the vicinity of 34°S.

Thank you, impact of the local wind forcing has now been investigated by estimating correlation between along-shore wind-stress and the cross-shelf transports at different locations following this article. It is found that away from the coast at about 1000m isobath local wind-stress does not influence the transport budget. This is now explained in p8, lines 4-18.

• *p* 20: The order of (*a*),...(*d*) at the top of the figure need checking?

Thank you, this figure is now corrected in the manuscript. (p23, figure 7).

**Issues of expression, grammar, etc**

*p* 2 line 5: "..long-term velocity observations" [there are other fisheries-biological data records and satellite SST that continue for many years/decades]

The text has been modified (p2, line 5).

*p* 2 line 9: insert of after study

The text has been modified (p2, line 9).

*p* 3 *line* 16: "... *two thirds* ... *contribute*..." Apologies, now corrected (p3, line 17).

*p* 4 line 8: "Further east ..." Shouldn't this be west? Thank you for pointing this out, now corrected (p4, line 9).

*p* 4 *line* 9: *conducted* Apologies, now corrected (p4, line 11).

*p* 4 *line* 17: "..*no long-term measurements of this current's flow/transport/dynamics.. are available*" The text has been modified (p4, line 20).

*p* 4 *line* 33: "0.5x0.5 *grid*": *It is suggested to change this to* 0.5°x0.5° *grid*" Sure, this has now been changed in the manuscript. (p5, line1)

*p* 5 *line* 8: "..*dynamic height does not change.*." Apologies, now corrected (p5, line 10).

*p* 5 *line* 15: *insert* "*and*" *after coast* Apologies, this paragraph has been rephrased.

*p* 5 *line* 26: *The convention is that positions should be reported as (lat, long) and, not as (long, lat).* Sure, we have corrected this in the manuscript (p6, line 1).

*p* 5 line 32. I don't see any "zonal velocities" in die area indicated by the authors. Do the authors mean "zonal velocity components" when they say "zonal velocity"? This was also mentioned above, and it is suggested that the authors rigorously identify and reword such misnomers.

Thank you for pointing this out. This has been taken care of (p6, line 6).

*p* 6 line 9. It is possible that Figure 2b shows a weak signal around 7°E corresponding to the third tributary referred to before (*p* 5 line 30).

Thank you, the manuscript has now been modified on this. (p 6, lines 7-9)

p 6 line 11-12: So the westward flow at 30°S is still the Benguela Current?

Yes; based on the definition of Benguela Current by Garzoli and Gordon (1996) at 30S the flow between the western edge of the Walvis ridge (at 3E) and the African coast is the Benguela Current.

*p* 6 line 20: The authors should indicate the origin of the standard deviation.

Standard deviation and mean were calculated for the entire time series between 1993 and 2015. This is now included in the text. (p 6, line 30)

p 6 line 32: transports

Thank you for pointing this out. This is now corrected. (p 7, line 4)

*p* 8 last paragraph: I liked the results concerning the course of Agulhas rings.

Thank you, we appreciate it.

p 9 line 4: cannot

Thank you. This is now corrected (p 10, line 15).

p 11 line 2: Vema

Thank you for catching the mistake. This is now corrected in the manuscript (p 12, line 11).

p16: "...the black straight line marks the depth.."

The text is corrected now (p 19, fig 3).